# A Unified Formulation of k-Means, Fuzzy c-Means and Gaussian Mixture Model by the Kolmogorov–Nagumo Average

**DOI:** 10.3390/e23050518

**Published:** 2021-04-24

**Authors:** Osamu Komori, Shinto Eguchi

**Affiliations:** 1Department of Computer and Information Science, Seikei University, 3-3-1 Kichijoji-Kitamachi, Musashino-shi, Tokyo 180-8633, Japan; 2The Institute of Statistical Mathematics, 10-3 Midori-cho, Tachikawa, Tokyo 190-8562, Japan; eguchi@ism.ac.jp

**Keywords:** k-means, fuzzy-c, Gaussian mixture model, Kolmogorov–Nagumo average, generalized energy function, Pareto distribution

## Abstract

Clustering is a major unsupervised learning algorithm and is widely applied in data mining and statistical data analyses. Typical examples include k-means, fuzzy c-means, and Gaussian mixture models, which are categorized into hard, soft, and model-based clusterings, respectively. We propose a new clustering, called Pareto clustering, based on the Kolmogorov–Nagumo average, which is defined by a survival function of the Pareto distribution. The proposed algorithm incorporates all the aforementioned clusterings plus maximum-entropy clustering. We introduce a probabilistic framework for the proposed method, in which the underlying distribution to give consistency is discussed. We build the minorize-maximization algorithm to estimate the parameters in Pareto clustering. We compare the performance with existing methods in simulation studies and in benchmark dataset analyses to demonstrate its highly practical utilities.

## 1. Introduction

In data analysis or data mining, there are two fundamental types of methodologies: clustering and classification [1]. Clustering, which is categorized as an exploratory paradigm, detects the underlying structure behind the data and grasps the rough image before proceeding to more intensive and comprehensive data analysis [2,3]. On the other hand, classification predicts unknown class labels of test data based on models constructed by training data with known class labels. The former is called supervised learning, while the latter is called unsupervised learning in pattern recognition [4].

Clustering algorithms fall roughly into three categories: hierarchical, partitioning, and mixture model-based algorithms [5]. In hierarchical clustering, each observation is considered as one cluster in the initial setting. Then clusters are merged recursively based on a similarity matrix defined beforehand. The resultant clusters are expressed as a dendrogram. The partition algorithm starts with a fixed number of clusters and searches for the cluster centers to minimize an objective function such as the squared distances between the centers and observations. It finds the centers simultaneously. The model-based algorithm assumes a mixture of probability distributions, which generates the observations and assigns the distributions to one of the mixture components. A Gaussian-mixture distribution-based approach is widely used in this context.

In this paper, we propose a new clustering, called Pareto clustering in the framework of quasilinear modeling [6,7,8]. It combines the cluster components by the Kolmogorov–Nagumo average [9] in a flexible way. We consider a generalized energy function as an objective function to estimate cluster parameters, which is an extension of the energy function proposed by [10]. The objective function consists of a survival function of the Pareto distribution, which is widely used in extreme value theory [11]. We investigate the consistency of the parameters, resulting in the underlying probability distribution of the generalized energy function. We find that k-means [12,13] and fuzzy c-means [14] have the underlying probability distributions with singular points at the cluster centers. This fact shows a clear difference from the model-based clustering such as a Gaussian-mixture modeling. Moreover, we show that the quasilinear modeling based on the Kolmogorov–Nagumo average connects k-means, fuzzy c-means, and a Gaussian-mixture modeling using the hyperparameters of the generalized energy function. See [15,16] for the discussion of the relation between k-means and fuzzy c-means.

The paper is organized as follows. In Section 2, we introduce the generalized energy function as the objective function of the Pareto clustering and discuss the consistency of the parameters. Moreover, we show that k-means, fuzzy c-means, and a Gaussian-mixture are all derived from the generalized energy function as special cases. This fact leads to the fact that the parameters can be estimated in a unified manner by the the minorize-maximization (MM) algorithm [17], where the monotone decrease of generalized energy function is guaranteed. In Section 3, we demonstrate the performance of the Pareto clustering based on simulation studies and benchmark datasets and show its practical utilities. We summarize the results of the Pareto clustering and discuss the extensions and applications in various scientific fields.

## 2. Materials and Methods

### 2.1. Generalized Energy Function

Let *T* be a non-negative random variable with a probability density function f(t). The survival function of *T* is defined as
(1)$$\begin{array}{c} S(t)=P(T>t),\;t\geq 0. \end{array}$$
Then for *d*-dimensional random variables $x_{1},\ldots ,x_{n}$, we define a generalized energy function to be minimized with respect to a parameter μ for clustering
(2)$$\begin{array}{ccc} L_{S}\left(\mu \right) & = & \frac{1}{\tau }\sum_{i=1}^{n}S^{-1}\left(\frac{1}{K}\sum_{k=1}^{K}S(\tau ∥x_{i}-\mu_{k}{∥}^{2})\right), \end{array}$$
where $\mu =(\mu_{1},\ldots ,\mu_{K})$ is a set of centers and τ>0 is the shape parameter. If we take S(t)=exp(t), the function corresponds to the energy function proposed by [10], where τ can be interpreted as the temperature in physics. The formulation in (Equation 2) is called the Kolmogorov–Nagumo average [9,18] and is widely applied to bioinformatics, ecology, fisheries, etc. [6,8,19].

In Equation (Equation 2), we express an average of probabilities that $x_{i}$ belongs to the *k*th cluster over all *K* clusters using $1/K\sum_{k=1}^{K}S(∥x_{i}-\mu_{k}{∥}^{2})$, where $∥x_{i}-\mu_{k}{∥}^{2}$ is the energy of $x_{i}$ associated with $\mu_{k}$. Hence we view $S^{-1}\left(1/K\sum_{k=1}^{K}S(∥x_{i}-\mu_{k}{∥}^{2})\right)$ as the Kolmogorov–Nagumo average of the energy of $x_{i}$ with the probabilistic meanings. In effect, we take summation of the Kolmogorov-Nagumo average over the observations $\left\{x_{1},...,x_{n}\right\}$.

**Remark** **1.***The generalized energy function* (2) *has a relation with the Archimedean copula defined by*
(3)$$\begin{array}{c} 1-S\left(\sum_{k=1}^{K}S^{-1}\left(1-u_{k}\right)\right) \end{array}$$
*for ${\left\{u_{k}\right\}}_{k=1}^{K}$ in (0,1), cf. [20] for an introductory discussion. In principle, the generalized energy function is a function from a vector of K cluster energy functions to a integrated energy function. The Archimedean copula is that of K marginal cumulative distribution functions to the joint cumulative distribution function. In this way, the generalized energy function expresses an interactive relation for cluster energy functions analogous with the Archimedean copula expressing the correlation among variables.*

We consider an estimator of the generalized energy function as
(4)$$\begin{array}{c} \hat{\mu }=\underset{\mu }{\mathrm{argmin}}L_{S}\left(\mu \right). \end{array}$$

If we assume that $x_{i}\;\left(i=1,\ldots ,n\right)$ is distributed according to a probability density function $p(x,\mu^{*})$, the expected generalized energy function is given by
(5)$$\begin{array}{c} L_{S}\left(\mu \right)=\frac{1}{\tau }\int S^{-1}\left(\frac{1}{K}\sum_{k=1}^{K}S(\tau ∥x-\mu_{k}{∥}^{2})\right)p\left(x,\mu^{*}\right)dx. \end{array}$$

Here we define a function for a set of cluster centers as
(6)$$\begin{array}{c} E_{\mu }\left(x\right)=\frac{1}{K}\sum_{k=1}^{K}S(\tau ∥x-\mu_{k}{∥}^{2}). \end{array}$$

Thus, we note that
(7)$$\begin{array}{c} \int E_{\mu }\left(x\right)dx=v_{d}E\left(T^{\frac{d}{2}}\right), \end{array}$$
where $v_{d}$ is a volume constant $2\pi^{d/2}/\left\{\tau^{1/2}d\Gamma \left(d/2\right)\right\}$ because $E\left(T\right)=\int_{0}^{\infty }S\left(t\right)dt$ (Appendix B). This property is a key idea in the following discussion.

**Lemma** **1.***Assume that the survival function S(t) in* (1) *is convex in t. We define a function G of $\left(\mu^{*},\mu \right)$ as*
(8)$$\begin{array}{c} G\left(\mu^{*},\mu \right)=\int S^{-1}\left(E_{\mu }\left(x\right)\right)f\left(S^{-1}\left(E_{\mu^{*}}\left(x\right)\right)\right)dx, \end{array}$$
*Then for any μ and $\mu^{*}$*
(9)$$\begin{array}{c} G\left(\mu^{*},\mu \right)\geq G\left(\mu^{*},\mu^{*}\right) \end{array}$$*with equality if and only if $\mu =\mu^{*}$.*

**Proof.** We observe that the function $S^{-1}\left(t\right)$ is a decreasing function of *t* given as
(10)$$\begin{array}{c} \frac{\partial S^{-1}\left(t\right)}{\partial t}=-\frac{1}{f(S^{-1}\left(t\right))} \end{array}$$
because $S(S^{-1}\left(t\right))=t$ and (∂/∂t)S(t)=−f(t). Similarly,
(11)$$\begin{array}{c} \frac{\partial^{2}S^{-1}\left(t\right)}{\partial t^{2}}=-\frac{f^{\prime }\left(S^{-1}\left(t\right)\right)}{{\left\{f\left(S^{-1}\left(t\right)\right)\right\}}^{3}}, \end{array}$$
which is positive for all t≥0 because $\left(\partial^{2}/\partial t^{2}\right)S\left(t\right)=-f^{\prime }\left(t\right)>0$ from the convexity assumption for S(t). Therefore, $S^{-1}\left(t\right)$ is also convex in t∈(0,1). This leads to
(12)$$\begin{array}{ccc} G\left(\mu^{*},\mu \right)-G\left(\mu^{*},\mu^{*}\right) & = & \int \left\{S^{-1}\left(E_{\mu }\left(x\right)\right)-S^{-1}\left(E_{\mu^{*}}\right)\right\}f\left(S^{-1}\left(E_{\mu^{*}}\left(x\right)\right)\right)dx \end{array}$$
(13)$$\begin{array}{ccc} & \geq & \int \left\{E_{\mu }\left(x\right)-E_{\mu^{*}}\left(x\right)\right\}\frac{\partial S^{-1}\left(E_{\mu^{*}}\left(x\right)\right)}{\partial t}f(S^{-1}\left(E_{\mu^{*}}\left(x\right)\right)dx \end{array}$$
(14)$$\begin{array}{ccc} & = & -\int \{E_{\mu }\left(x\right)-E_{\mu^{*}}\left(x\right)\}dx \end{array}$$
(15)$$\begin{array}{ccc} & = & 0. \end{array}$$Here Equality in (13) holds if and only if $\mu =\mu^{*}$ from the convexity for $S^{-1}$. The Equality (14) is shown by
(16)$$\begin{array}{c} -f\left(S^{-1}\left(t\right)\right)\frac{\partial S^{-1}\left(t\right)}{\partial t}=1 \end{array}$$
for any t≥0 as seen in (10). Equality (15) holds due to (7).    □

**Theorem** **1.**
*If the $p(x,\mu^{*})$ has a form such as*
(17)$$\begin{array}{c} p\left(x,\mu^{*}\right)=Z\left(\mu^{*}\right)f\left(S^{-1}\left(E_{\mu^{*}}\left(x\right)\right)\right), \end{array}$$
*where $Z(\mu^{*})>0$ is a normalizing constant. Then we have*
(18)$$\begin{array}{c} L_{S}\left(\mu \right)\geq L_{S}\left(\mu^{*}\right). \end{array}$$


**Proof.** Note that
$$\begin{array}{ccc} L_{S}\left(\mu \right)-L_{S}\left(\mu^{*}\right) & = & \frac{Z(\mu^{*})}{\tau }\left\{G\left(\mu^{*},\mu \right)-G\left(\mu^{*},\mu^{*}\right)\right\}, \end{array}$$
which concludes (Equation 18) from Lemma 1.    □

We note that $\hat{\mu }$ is asymptotically consistent to true parameter $\mu^{*}$ if the probability density function has the form in (Equation 17).

#### 2.1.1. Pareto Distribution

Let us consider a generalized Pareto distribution, where the survival function and its inverse function are defined by
$$\begin{array}{c} S\left(t\right)={\left(1+\beta t\right)}^{-\frac{1}{\beta }}\;\;\mathrm{and}\;\;S^{-1}\left(t\right)=\frac{t^{-\beta }-1}{\beta }, \end{array}$$
where β>0 denotes the shape parameter. Then the generalized energy function is
(19)$$\begin{array}{ccc} L_{\tau ,\beta }\left(\mu \right) & = & \frac{1}{\tau \beta }\sum_{i=1}^{n}\left[{\left\{\sum_{k=1}^{K}\frac{1}{K}\{1+\tau \beta ∥x_{i}-\mu_{k}{{∥}^{2}\}}^{-\frac{1}{\beta }}\right\}}^{-\beta }-1\right]. \end{array}$$
If we consider β→0, then
(20)$$\begin{array}{c} \lim_{\beta \to 0}L_{\tau ,\beta }\left(\mu \right)=-\frac{1}{\tau }\sum_{i=1}^{n}\log\left\{\sum_{k=1}^{K}\frac{1}{K}\exp\left(-\tau ∥x_{i}-\mu_{k}{∥}^{2}\right)\right\}, \end{array}$$
which is reduced to the energy function proposed by [10]. Hence, we can understand that Rose’s clustering (maximum-entropy clustering) is generated by a survival distribution function of an exponential distribution. Then we have
(21)$$\begin{array}{cc} \lim_{\tau \to \infty }L_{\tau ,\beta }\left(\mu \right)= & \lim_{\tau \to \infty }\frac{1}{\tau \beta }\sum_{i=1}^{n}{\left\{\sum_{k=1}^{K}\frac{1}{K}\{1+\tau \beta ∥x_{i}-\mu_{k}{{∥}^{2}\}}^{-\frac{1}{\beta }}\right\}}^{-\beta } \end{array}$$
(22)$$\begin{array}{cc} = & \sum_{i=1}^{n}{\left\{\sum_{k=1}^{K}\frac{1}{K}\{∥x_{i}-\mu_{k}{{∥}^{2}\}}^{-\frac{1}{\beta }}\right\}}^{-\beta } \end{array}$$
The gradient with respect to $\mu_{k}$ is given by
(23)$$\begin{array}{c} \frac{2}{\beta }\sum_{i=1}^{n}\frac{1}{K}{\left[\frac{\{∥x_{i}-\mu_{k}{{∥}^{2}\}}^{-\frac{1}{\beta }}}{\sum_{\ell =1}^{K}\pi_{\ell }\{∥x_{i}-\mu_{\ell }{{∥}^{2}\}}^{-\frac{1}{\beta }}}\right]}^{1+\beta }\left(x_{i}-\mu_{k}\right), \end{array}$$
which exactly leads to the estimation equations of fuzzy c-means if we take β=m−1 [14]. Furthermore, we have
(24)$$\begin{array}{ccc} \lim_{\tau \to \infty ,\beta \to 0}L_{\tau ,\beta }\left(\mu \right) & = & \sum_{i=1}^{n}\min_{1\leq k\leq K}{∥x_{i}-\mu_{k}∥}^{2}, \end{array}$$
which is the loss function of k-means. The corresponding survival function is $\lim_{\beta \to 0}{\left(1+\beta t\right)}^{-\frac{1}{\beta }}=\exp\left(-t\right)$. Note that the loss function is directly derived from (Equation 2) as
(25)$$\lim_{\tau \to \infty }\frac{1}{\tau }\sum_{i=1}^{n}S^{-1}\left(\frac{1}{K}\sum_{k=1}^{K}S(\tau ∥x_{i}-\mu_{k}{∥}^{2})\right)=\sum_{i=1}^{n}\min_{1\leq k\leq K}{∥x_{i}-\mu_{k}∥}^{2}.$$

In addition, we have
(26)$$\lim_{\tau \to 0}\frac{1}{\tau }\sum_{i=1}^{n}S^{-1}\left(\frac{1}{K}\sum_{k=1}^{K}S(\tau ∥x_{i}-\mu_{k}{∥}^{2})\right)=\sum_{i=1}^{n}\frac{1}{K}\sum_{k=1}^{K}{∥x_{i}-\mu_{k}∥}^{2}$$
because S(0)=1.

#### 2.1.2. Fréchet Distribution

Next, we consider Fréchet distribution with the survival function defined as
(27)$$\begin{array}{c} S\left(t\right)=1-\exp\left(-t^{\gamma }\right). \end{array}$$
where γ<0 is the shape parameter. The generalized energy function is given by
(28)$$\begin{array}{ccc} L_{\gamma ,\tau }\left(\mu \right) & & =\sum_{i=1}^{n}{\left(-\frac{1}{\tau^{\gamma }}\log\left[\frac{1}{K}\sum_{k=1}^{K}\exp(-\tau^{\gamma }∥x_{i}-\mu_{k}{∥}^{2\gamma })\right]\right)}^{\frac{1}{\gamma }}. \end{array}$$

We find that
(29)$$\begin{array}{ccc} \lim_{\tau \to 0}L_{\gamma ,\tau }\left(\mu \right) & = & \sum_{i=1}^{n}\lim_{\tau \to 0}{\left(-\frac{1}{\tau^{\gamma }}\log\left[\frac{1}{K}\sum_{k=1}^{K}\exp(-\tau^{\gamma }∥x_{i}-\mu_{k}{∥}^{2\gamma })\right]\right)}^{\frac{1}{\gamma }}\\ & = & \sum_{i=1}^{n}{\left(\min_{1\leq k\leq K}{∥x_{i}-\mu_{k}∥}^{2\gamma }\right)}^{\frac{1}{\gamma }}\\ & = & \sum_{i=1}^{n}\min_{1\leq k\leq K}{∥x_{i}-\mu_{k}∥}^{2}. \end{array}$$
Hence, this energy function is reduced to that of the *K*-means algorithm as shown in the Pareto distribution case. The estimating equation is given by
(30)$$\begin{array}{ccc} \frac{\partial }{\partial \mu_{k}}L_{\gamma ,\tau }\left(\mu \right) & = & \sum_{i=1}^{n}\omega_{k}\left(x_{i},\tau ,\gamma \right)\left(\mu_{k}-x_{i}\right)=0, \end{array}$$
where
(31)$$\begin{array}{ccc} \omega_{k}\left(x_{i},\tau ,\gamma \right) & = & {\left(-\frac{1}{\tau^{\gamma }}\log\left[\frac{1}{K}\sum_{\ell =1}^{K}\exp(-\tau^{\gamma }∥x_{i}-\mu_{\ell }{∥}^{2\gamma })\right]\right)}^{\frac{1}{\gamma }-1}\\ & & \times \frac{\exp(-\tau^{\gamma }∥x_{i}-\mu_{k}{∥}^{2\gamma })}{\sum_{\ell =1}^{K}\exp(-\tau^{\gamma }∥x_{i}-\mu_{\ell }{∥}^{2\gamma })}{∥x_{i}-\mu_{k}∥}^{2\gamma -2}. \end{array}$$
When we assume the unbiasedness for the estimating function in (30), that is
(32)$$\begin{array}{c} E\{\omega_{k}\left(X,\tau ,\gamma \right)\left(\mu_{k}-X\right)\}=0, \end{array}$$
the underlying distribution has a density function proportional to
(33)$$\begin{array}{c} {\left(-\frac{1}{\tau^{\gamma }}\log M\left(x,\mu \right)\right)}^{\frac{\gamma }{1-\gamma }}M\left(x,\mu \right). \end{array}$$
where
(34)$$\begin{array}{c} M\left(x,\mu \right)=\frac{1}{K}\sum_{\ell =1}^{K}\exp(-\tau^{\gamma }∥x-\mu_{\ell }{∥}^{2\gamma }). \end{array}$$

We confirm that
(35)$$\begin{array}{c} \omega_{k}\left(x_{i},\tau ,\gamma \right)=\left\{\begin{array}{cc} 1 & \mathrm{if}\;∥x_{i}-\mu_{k}{∥}^{2}=\min_{1\leq \ell \leq K}{∥x_{i}-\mu_{\ell }∥}^{2}\\ 0 & \mathrm{otherwise} \end{array}\right. \end{array}$$
as τ goes to 0. Then we consider the limit of τ to *∞*, which provides
(36)$$\begin{array}{c} \lim_{\tau \to \infty }L_{\gamma ,\tau }\left(\mu \right)=\sum_{i=1}^{n}\lim_{\tau \to \infty }{\left(-\frac{1}{\tau^{\gamma }}\log\left[\frac{1}{K}\sum_{k=1}^{K}\exp(-\tau^{\gamma }∥x_{i}-\mu_{k}{∥}^{2\gamma })\right]\right)}^{\frac{1}{\gamma }}\\ \;\;\;\;\;\;\;\;\;\;\;\;\;\;\;\;\;\;\;\;\;\;\;\;\;\;\;=\sum_{i=1}^{n}{\left(\frac{1}{K}\sum_{k=1}^{K}{∥x_{i}-\mu_{k}∥}^{2\gamma }\right)}^{\frac{1}{\gamma }} \end{array}$$
which is equal to (22). This also leads to Fuzzy *c*-means if we take as γ=1/(1−m) [14].

### 2.2. Estimation of Variances and Mixing Proportions in Clusters

In stead of the Euclidean distance $∥x_{i}-\mu_{k}{∥}^{2}$, we consider $∥x_{i}-\mu_{k}{∥}_{\sum_{k}^{-1}}^{2}={\left(x_{i}-\mu_{k}\right)}^{⊤}\sum_{k}^{-1}\left(x_{i}-\mu_{k}\right)$ to incorporate the variance structure around $\mu_{k}$. Bezdek et al. [14] considered a common variance structure $\sum_{k}=\sum_{i=1}^{n}\left(x_{i}-\bar{x}\right){\left(x_{i}-\bar{x}\right)}^{⊤}$, where $\bar{x}=1/n\sum_{i=1}^{n}x_{i}$ for k=1,…,K. On the other hand, we estimate distinct $\sum_{k}$ for each $\mu_{k}$.

For this purpose, we modify the generalized energy function in (Equation 2) to allow for a variances $\sum_{1},\ldots ,\sum_{K}$ and mixing proportions $\pi_{1},\ldots ,\pi_{K}$ ($\sum_{k=1}^{K}\pi_{k}=1$ and $\pi_{k}\geq 0$ for k=1,…,K) as
(37)$$\begin{array}{ccc} L_{S}\left(\theta \right) & = & \frac{1}{\tau }\sum_{i=1}^{n}S^{-1}\left(\sum_{k=1}^{K}\pi_{k}|\sum_{k}{|}^{-\frac{1}{2}}S(\tau ∥x_{i}-\mu_{k}{∥}_{\sum_{k}^{-1}}^{2})\right), \end{array}$$
where $\theta ={\left(\mu_{k},\sum_{k},\pi_{k}\right)}_{k=1}^{K}$. We assume that S(t) is convex so that the domain of $S^{-1}\left(t\right)$ can be extended from [0,1] to [0,∞) to allow for $|\sum_{k}{|}^{-\frac{1}{2}}$. The estimator of this modified generalized energy function is given as
(38)$$\begin{array}{c} \hat{\theta }=\underset{\theta }{\mathrm{argmin}}L_{S}\left(\theta \right), \end{array}$$
The expected generalized energy function is given by
(39)$$\begin{array}{c} L_{S}\left(\theta \right)=\frac{1}{\tau }\int S^{-1}\left(\sum_{k=1}^{K}\pi_{k}|\sum_{k}{|}^{-\frac{1}{2}}S(\tau ∥x-\mu_{k}{∥}_{\sum_{k}^{-1}}^{2})\right)p\left(x,\theta^{*}\right)dx, \end{array}$$
where $p(x,\theta^{*})$ is the underlying probability density function.

For a cumulative distribution function F(t)=1−S(t) we have $L_{S}\left(\theta \right)=L_{F}\left(\theta \right)$ if and only if $\sum_{k=1}^{K}\pi_{k}{\left|\sum_{k}\right|}^{-1/2}=1$. On the other hand, it always holds that $L_{S}\left(\mu \right)=L_{F}\left(\mu \right)$ for the original generalized energy function in (Equation 2).

Similarly to (Equation 6), we define
(40)$$\begin{array}{c} E_{\theta }\left(x\right)=\sum_{k=1}^{K}\pi_{k}|\sum_{k}{|}^{-\frac{1}{2}}S(\tau ∥x-\mu_{k}{∥}_{\sum_{k}^{-1}}^{2}), \end{array}$$
and we notice that $\int E_{\theta }\left(x\right)dx$ is also independent of μ as
(41)$$\int E_{\theta }\left(x\right)dx=v_{d}E\left(T^{\frac{d}{2}}\right).$$

**Lemma** **2.***Assume that the survival function S(t) in* (1) *is convex in t. We define a function G of $\left(\mu^{*},\mu \right)$ as*
(42)$$\begin{array}{c} G\left(\theta^{*},\theta \right)=\int S^{-1}\left(E_{\theta }\left(x\right)\right)f\left(S^{-1}\left(E_{\theta^{*}}\left(x\right)\right)\right)dx. \end{array}$$
*Then for any θ and $\theta^{*}$*
(43)$$\begin{array}{c} G\left(\theta^{*},\theta \right)\geq G\left(\theta^{*},\theta^{*}\right) \end{array}$$*with equality if and only if $\theta =\theta^{*}$.*

**Proof.** It is obvious from Lemma 1 and the fact that $\int E_{\theta }\left(x\right)dx$ is independent of μ.    □

From Lemma 2, we can easily show the following theorem regarding $L_{S}\left(\theta \right)$.

**Theorem** **2.**
*If the $p(x,\theta^{*})$ has a form such as*
(44)$$\begin{array}{c} p\left(x,\theta^{*}\right)=Z\left(\theta^{*}\right)f\left(S^{-1}\left(E_{\theta^{*}}\left(x\right)\right)\right), \end{array}$$
*where $Z(\theta^{*})>0$ is a normalizing constant. Then we have*
(45)$$\begin{array}{c} L_{S}\left(\theta \right)\geq L_{S}\left(\theta^{*}\right). \end{array}$$


For the Pareto distribution, we have from (Equation 37)
(46)$$\begin{array}{ccc} L_{\tau ,\beta }\left(\theta \right) & = & \frac{1}{\tau \beta }\sum_{i=1}^{n}\left[{\left\{\sum_{k=1}^{K}\pi_{k}|\sum_{k}{|}^{-\frac{1}{2}}\{1+\tau \beta ∥x_{i}-\mu_{k}{{∥}_{\sum_{k}^{-1}}^{2}\}}^{-\frac{1}{\beta }}\right\}}^{-\beta }-1\right] \end{array}$$
(47)$$\begin{array}{ccc} & = & \frac{1}{\tau }\sum_{i=1}^{n}\phi \left(\sum_{k=1}^{K}\pi_{k}w\left(x_{i},\mu_{k},\sum_{k}\right)\right), \end{array}$$
where
(48)$$\begin{array}{ccc} w(x_{i},\mu_{k},\sum_{k}) & = & |\sum_{k}{|}^{-\frac{1}{2}}\{1+\tau \beta ∥x_{i}-\mu_{k}{{∥}_{\sum_{k}^{-1}}^{2}\}}^{-\frac{1}{\beta }} \end{array}$$
(49)$$\begin{array}{ccc} \phi (t) & = & \frac{t^{-\beta }-1}{\beta }. \end{array}$$
From (Equation 44), the underlying probability density function is
(50)$$p_{\tau ,\beta }\left(\theta^{*}\right)=Z_{\tau ,\beta }\left(\theta^{*}\right){\left\{\sum_{k=1}^{K}\pi_{k}^{*}w\left(x_{i},\mu_{k}^{*},\sum_{k}^{*}\right)\right\}}^{1+\beta },$$
where $Z_{\tau ,\beta }\left(\theta^{*}\right)$ is a normalizing constant. When β→0, we have
(51)$$\lim_{\beta \to 0}L_{\tau ,\beta }\left(\theta \right)=-\frac{1}{\tau }\sum_{i=1}^{n}\log\left\{\sum_{k=1}^{K}\pi_{k}|\sum_{k}{|}^{-\frac{1}{2}}\exp(-\tau ∥x_{i}-\mu_{k}{∥}_{\sum_{k}^{-1}}^{2})\right\},$$
which is the negative log likelihood function of the normal mixture distributions apart from a constant term ${\left(2\pi \right)}^{-d/2}$ when τ=1/2.

Similarly, we have the estimation equation of fuzzy c-means allowing for the Mahalanobis distance when τ→∞. Moreover, we have k-means with the use of the Mahalanobis distance when τ→∞ and β→0. For the other extreme cases, we observe that both $\lim_{\beta \to \infty }L_{\tau ,\beta }\left(\theta \right)$ and $\lim_{\tau \to 0}L_{\tau ,\beta }\left(\theta \right)$ diverge or converge to 0 depending on the values of $\pi_{k}$ and $\sum_{k}$ (k=1,…,K). Hence we choose large values for τ and small values for β in the subsequent data analysis.

### 2.3. Estimating Algorithm

The direct optimization of $L_{\tau ,\beta }\left(\theta \right)$ in (Equation 46) is difficult due to the mixture structure. Thus, we employ the idea of expectation and maximization (EM) algorithm [21] and the minorize-maximization (MM) algorithm [17] similar to [19]. Our proposed clustering method (Pareto clustering) is as follows in Algorithm 1.
**Algorithm 1:** Pareto clustering1. Set initial values $\left(\mu_{k}^{\left(0\right)},\sum_{k}^{\left(0\right)},\pi_{k}^{\left(0\right)}\right)$ for k=1,…,K. 
2. Repeat the following steps for t=0,…,T−1 and k=1,…,K until convergence. 
3. (52)$$\begin{array}{c} q_{k}^{\left(t\right)}\left(x_{i}\right)=\frac{\pi_{k}^{\left(t\right)}w\left(x_{i},\mu_{k}^{\left(t\right)},\sum_{k}^{\left(t\right)}\right)}{\sum_{\ell =1}^{K}\pi_{k}^{\left(t\right)}w\left(x_{i},\mu_{k}^{\left(t\right)},\sum_{k}^{\left(t\right)}\right)} \end{array}$$
(53)$$\begin{array}{c} \mu_{k}^{\left(t+1\right)}=\frac{\sum_{i=1}^{n}{\left\{q_{k}^{\left(t\right)}\left(x_{i}\right)\right\}}^{1+\beta }x_{i}}{\sum_{i=1}^{n}{\left\{q_{k}^{\left(t\right)}\left(x_{i}\right)\right\}}^{1+\beta }} \end{array}$$(54)$$\sum_{k}^{\left(t+1\right)}=\frac{\tau \left(2-d\beta \right)\sum_{i=1}^{n}{\left\{q_{k}^{\left(t\right)}\left(x_{i}\right)\right\}}^{1+\beta }\left(x_{i}-\mu_{k}^{\left(t+1\right)}\right){\left(x_{i}-\mu_{k}^{\left(t+1\right)}\right)}^{⊤}}{\sum_{i=1}^{n}{\left\{q_{k}^{\left(t\right)}\left(x_{i}\right)\right\}}^{1+\beta }}$$(55)$$\pi_{k}^{\left(t+1\right)}=\frac{{\left\{\sum_{i=1}^{n}{\left\{q_{k}^{\left(t\right)}\left(x_{i}\right)\right\}}^{1+\beta }{\left\{w(x_{i},\mu_{k}^{\left(t+1\right)},\sum_{k}^{\left(t+1\right)})\right\}}^{-\beta }\right\}}^{\frac{1}{1+\beta }}}{\sum_{\ell =1}^{K}{\left\{\sum_{i=1}^{n}{\left\{q_{\ell }^{\left(t\right)}\left(x_{i}\right)\right\}}^{1+\beta }{\left\{w(x_{i},\mu_{\ell }^{\left(t+1\right)},\sum_{\ell }^{\left(t+1\right)})\right\}}^{-\beta }\right\}}^{\frac{1}{1+\beta }}}$$ 4. Output $\left({\hat{\mu }}_{k},{\hat{\sum }}_{k},{\hat{\pi }}_{k}\right)=\left(\mu_{k}^{\left(T\right)},\sum_{k}^{\left(T\right)},\pi_{k}^{\left(T\right)}\right)$ for k=1,…,K. 


The initial values $\left(\mu_{k}^{\left(0\right)},\sum_{k}^{\left(0\right)},\pi_{k}^{\left(0\right)}\right)$ are determined by the hierarchical clustering in a similar way to the algorithm by [22]. The derivation of the estimating algorithm is as follows. First, we have
(56)$$\begin{array}{ccc} L_{\tau ,\beta }\left(\theta \right) & = & \frac{1}{\tau }\sum_{i=1}^{n}\phi \left(\sum_{k=1}^{K}\pi_{k}w\left(x_{i},\mu_{k},\sum_{k}\right)\right) \end{array}$$
(57)$$\begin{array}{ccc} & = & \frac{1}{\tau }\sum_{i=1}^{n}\phi \left(\sum_{k=1}^{K}q_{k}\left(x_{i}\right)\frac{\pi_{k}w\left(x_{i},\mu_{k},\sum_{k}\right)}{q_{k}\left(x_{i}\right)}\right) \end{array}$$
(58)$$\begin{array}{ccc} & \leq & \frac{1}{\tau }\sum_{i=1}^{n}\sum_{k=1}^{K}q_{k}\left(x_{i}\right)\phi \left(\frac{\pi_{k}w\left(x_{i},\mu_{k},\sum_{k}\right)}{q_{k}\left(x_{i}\right)}\right) \end{array}$$
where $q_{k}\left(x_{i}\right)$ is a positive weight such as $\sum_{k=1}^{K}q_{k}\left(x_{i}\right)=1$ and ϕ(t) is the convex function defined in (49). The equality holds if and only if
(59)$$\begin{array}{c} \frac{\pi_{1}w\left(x_{i},\mu_{1},\sum_{1}\right)}{q_{1}\left(x_{i}\right)}=\cdots =\frac{\pi_{K}w\left(x_{i},\mu_{K},\sum_{K}\right)}{q_{K}\left(x_{i}\right)}, \end{array}$$
which is equivalent to
(60)$$\begin{array}{c} q_{k}\left(x_{i}\right)=\frac{\pi_{k}w\left(x_{i},\mu_{k},\sum_{k}\right)}{\sum_{k=1}^{K}\pi_{k}w\left(x_{i},\mu_{k},\sum_{k}\right)},\;\left(k=1,\ldots ,K\right). \end{array}$$
Based on $q_{k}^{\left(t\right)}\left(x_{i}\right)$ in (Equation 52), we define
(61)$$\begin{array}{cc} Q(\theta |\theta^{\left(t\right)})= & \frac{1}{\tau }\sum_{i=1}^{n}\sum_{k=1}^{K}q_{k}^{\left(t\right)}\left(x_{i}\right)\left\{{\left(\frac{\pi_{k}w\left(x_{i},\mu_{k},\sum_{k}\right)}{q_{k}^{\left(t\right)}\left(x_{i}\right)}\right)}^{-\beta }-1\right\}/\beta \end{array}$$
(62)$$\begin{array}{cc} = & \frac{1}{\tau \beta }\sum_{i=1}^{n}\sum_{k=1}^{K}\left[{\left\{q_{k}^{\left(t\right)}\left(x_{i}\right)\right\}}^{1+\beta }\pi_{k}^{-\beta }{\left|\sum_{k}\right|}^{\frac{\beta }{2}}\left\{1+\tau \beta {\left(x_{i}-\mu_{k}\right)}^{⊤}\sum_{k}^{-1}\left(x_{i}-\mu_{k}\right)\right\}-q_{k}^{\left(t\right)}\left(x_{i}\right)\right] \end{array}$$
(63)$$\begin{array}{cc} = & \frac{1}{\tau \beta }\sum_{i=1}^{n}\sum_{k=1}^{K}\left[{\left\{q_{k}^{\left(t\right)}\left(x_{i}\right)\right\}}^{1+\beta }\pi_{k}^{-\beta }\left\{\right|V_{k}^{-1}{|}^{\frac{\beta }{d\beta -2}}+\tau \beta {\left(x_{i}-\mu_{k}\right)}^{⊤}V_{k}^{-1}\left(x_{i}-\mu_{k}\right)\}-q_{k}^{\left(t\right)}\left(x_{i}\right)\right], \end{array}$$
where $V_{k}^{-1}={\left|\sum_{k}\right|}^{\beta /2}\sum_{k}^{-1}$. Then we have
(64)$$\begin{array}{ccc} \frac{\partial }{\partial \mu_{k}}Q\left(\theta |\theta^{\left(t\right)}\right) & = & -2\pi_{k}^{-\beta }V_{k}^{-1}\sum_{i=1}^{n}{\left\{q_{k}^{\left(t\right)}\left(x_{i}\right)\right\}}^{1+\beta }\left(x_{i}-\mu_{k}\right) \end{array}$$
(65)$$\begin{array}{ccc} & = & 0 \end{array}$$
which means that
(66)$$\begin{array}{c} \mu_{k}^{\left(t+1\right)}=\frac{\sum_{i=1}^{n}{\left\{q_{k}^{\left(t\right)}\left(x_{i}\right)\right\}}^{1+\beta }x_{i}}{\sum_{i=1}^{n}{\left\{q_{k}^{\left(t\right)}\left(x_{i}\right)\right\}}^{1+\beta }}. \end{array}$$
Similarly, we have
(67)$$\begin{array}{cc} \frac{\partial }{\partial V_{k}^{-1}}Q\left(\theta |\theta^{\left(t\right)}\right){|}_{\mu_{k}=\mu_{k}^{\left(t+1\right)}}\; & =\frac{1}{\tau \beta }\sum_{i=1}^{n}{\left\{q_{k}^{\left(t\right)}\left(x_{i}\right)\right\}}^{1+\beta }\pi_{k}^{-\beta }\left\{\frac{\beta }{d\beta -2}{\left|V_{k}^{-1}\right|}^{\frac{\beta }{d\beta -2}}V_{k}+\tau \beta \left(x_{i}-\mu_{k}^{\left(t+1\right)}\right){\left(x_{i}-\mu_{k}^{\left(t+1\right)}\right)}^{⊤}\right\} \end{array}$$
(68)$$\begin{array}{cc} \; & =\frac{1}{\tau \beta }\sum_{i=1}^{n}{\left\{q_{k}^{\left(t\right)}\left(x_{i}\right)\right\}}^{1+\beta }\pi_{k}^{-\beta }\left\{\frac{\beta }{d\beta -2}\sum_{k}+\tau \beta \left(x_{i}-\mu_{k}^{\left(t+1\right)}\right){\left(x_{i}-\mu_{k}^{\left(t+1\right)}\right)}^{⊤}\right\} \end{array}$$
(69)$$\begin{array}{cc} \; & =0, \end{array}$$
which means that
(70)$$\begin{array}{c} \sum_{k}^{\left(t+1\right)}=\frac{\tau \left(2-d\beta \right)\sum_{i=1}^{n}{\left\{q_{k}^{\left(t\right)}\left(x_{i}\right)\right\}}^{1+\beta }\left(x_{i}-\mu_{k}^{\left(t+1\right)}\right){\left(x_{i}-\mu_{k}^{\left(t+1\right)}\right)}^{⊤}}{\sum_{i=1}^{n}{\left\{q_{k}^{\left(t\right)}\left(x_{i}\right)\right\}}^{1+\beta }}. \end{array}$$
Next we consider
(71)$$\begin{array}{c} R\left(\theta |\theta^{\left(t\right)}\right)=Q\left(\theta |\theta^{\left(t\right)}\right)+\lambda \left(\sum_{k=1}^{K}\pi_{k}-1\right), \end{array}$$
where λ is a Lagrange multiplier. Then
(72)$$\begin{array}{ccc} \frac{\partial }{\partial \pi_{k}}R\left(\theta |\theta^{\left(t\right)}\right){|}_{\mu_{k}=\mu_{k}^{\left(t+1\right)},\sum_{k}=\sum_{k}^{\left(t+1\right)}} & = & -\frac{1}{\tau }\pi_{k}^{-\beta -1}\sum_{i=1}^{n}{\left\{q_{k}^{\left(t\right)}\left(x_{i}\right)\right\}}^{1+\beta }w{\left(x_{i},\mu_{k}^{\left(t+1\right)},\sum_{k}^{\left(t+1\right)}\right)}^{-\beta }+\lambda \end{array}$$
(73)$$\begin{array}{ccc} & = & 0, \end{array}$$
which means
(74)$$\begin{array}{ccc} \pi_{k}^{\left(t+1\right)} & = & \frac{{\left\{\sum_{i=1}^{n}{\left\{q_{k}^{\left(t\right)}\left(x_{i}\right)\right\}}^{1+\beta }w{\left(x_{i},\mu_{k}^{\left(t+1\right)},\sum_{k}^{\left(t+1\right)}\right)}^{-\beta }\right\}}^{\frac{1}{1+\beta }}}{\sum_{\ell =1}^{K}{\left\{\sum_{i=1}^{n}{\left\{q_{\ell }^{\left(t\right)}\left(x_{i}\right)\right\}}^{1+\beta }w{\left(x_{i},\mu_{\ell }^{\left(t+1\right)},\sum_{\ell }^{\left(t+1\right)}\right)}^{-\beta }\right\}}^{\frac{1}{1+\beta }}}. \end{array}$$

**Remark** **2.**
*The generalized energy function in (Equation 46) is monotonically decreasing in the estimating algorithm. That is, we have*
(75)$$L_{\tau ,\beta }\left(\theta^{\left(t+1\right)}\right)\leq Q\left(\theta^{\left(t+1\right)}|\theta^{\left(t\right)}\right)\leq Q\left(\theta^{\left(t\right)}|\theta^{\left(t\right)}\right)=L_{\tau ,\beta }\left(\theta^{\left(t\right)}\right).$$
*See Appendix C for more details.*


**Remark** **3.**
*The estimating algorithm of fuzzy c-means by [14] is given as*
(76)$$\begin{array}{ccc} u_{ik}^{\left(t\right)} & = & {\left[\sum_{\ell =1}^{K}{\left(\frac{∥x_{i}-\mu_{k}^{\left(t\right)}{∥}^{2}}{∥x_{i}-\mu_{\ell }^{\left(t\right)}{∥}^{2}}\right)}^{\frac{1}{m-1}}\right]}^{-m}, \end{array}$$
(77)$$\begin{array}{ccc} \mu_{k}^{\left(t\right)} & = & \frac{\sum_{i=1}^{n}u_{ik}^{\left(t\right)}x_{i}}{\sum_{i=1}^{n}u_{ik}^{\left(t\right)}}, \end{array}$$
*where ${\left\{u_{ik}^{\left(t\right)}\right\}}^{1/m}$ is called the membership function of $x_{i}$ in cluster k at the iteration step t. These are special cases of (Equation 52) and (Equation 53) with τ→∞, $\sum_{k}=I$, $\pi_{k}=1/K$ and β=m−1. Hence we observe that the original algorithm of fuzzy c-means can be interpreted as the EM algorithm.*


**Remark** **4.**
*In analogy with the membership function of fuzzy c-means by [14], we define $q_{k}^{\left(t\right)}\left(x_{i}\right)$ in (Equation 52) as a membership function of $x_{i}$ in cluster k at the iteration step t in Pareto clustering. Hence we estimate cluster $C_{k}$ as*
(78)$$C_{k}=\left\{x_{i}|q_{k}^{\left(T\right)}\left(x_{i}\right)\geq q_{\ell }^{\left(T\right)}\left(x_{i}\right),\ell =1,\ldots ,K,\;i=1,\ldots ,n\right\},$$
*where $\cup_{k=1}^{K}C_{k}=\left\{x_{1},\ldots ,x_{n}\right\}$.*


**Remark** **5.**
*In high-dimensional setting p≫1, we consider the ridge regularization for $\sum_{k}^{\left(t+1\right)}$ as in [23]*
$$\sum_{k}^{\left(t+1\right)}\left(\alpha \right)=\alpha \sum_{k}^{\left(t+1\right)}+\left(1-\alpha \right){\hat{\sigma }}_{k}^{2}I,$$
*where α=0.95 and $\sigma_{k}^{2}$ is the scalar variance estimated to be the maximum value of the diagonal elements of $\sum_{k}^{\left(t+1\right)}$. Moreover, we take β≪1 to make $\sum_{k}^{\left(t+1\right)}$ positive definite.*


### 2.4. Evaluation of Clustering Methods

We compare the performances of k-means, fuzzy c-means, Gaussian mixture modeling (Gaussian), partitioning around medoids (PAM), and Pareto clustering. To implement these methods, we use the kmeans function in the stat package [24], the cmeans function in the e1071 package [14], the Mclust in the mclust package [22] and the pam function in the cluster package [25] in the statistical software R, where the default settings are used for each function. In Pareto clustering, τ=0.5 and β=1 are used as the default settings. We assume that the number of clusters *K* is known and compare the performances as in [26].

#### 2.4.1. Metrics

Cluster $C_{k}$ (k=1,…,K) estimated by a clustering method is evaluated by a predefined reference class set $D_{\ell }$ (ℓ=1,…,L) such as
$$\begin{array}{ccc} \mathrm{Precision}(C_{k},D_{\ell }) & = & \frac{|C_{k}\cap D_{\ell }|}{|C_{k}|}\\ \mathrm{Recall}(C_{k},D_{\ell }) & = & \frac{|C_{k}\cap D_{\ell }|}{|D_{\ell }|}, \end{array}$$
where $\mathrm{Recall}\left(C_{k},D_{\ell }\right)=\mathrm{Precision}\left(D_{\ell },C_{k}\right)$. $\mathrm{Precision}(C_{k},D_{\ell })$ counts data points in cluster $C_{k}$ belonging to class *ℓ*. Hence $\max_{\ell }\mathrm{Precision}\left(C_{k},D_{\ell }\right)$ represents the purity of cluster $C_{k}$ regarding the classes. By taking the weighted average, we have
$$\mathrm{Purity}=\sum_{k=1}^{K}\frac{|C_{k}|}{n}\max_{\ell }\mathrm{Precision}\left(C_{k},D_{\ell }\right),$$
where *n* is the sample size. $\mathrm{Recall}(C_{k},D_{\ell })$ counts data points in a class set $D_{\ell }$ estimated to be in cluster $C_{k}$. Precision and recall correspond to the positive predictive value and sensitivity, respectively [27].

A metric combining precision and recall is proposed by [28] such as
$$\text{F-value}=\sum_{k=1}^{K}\frac{|D_{k}|}{n}\max_{\ell }F\left(D_{k},C_{\ell }\right),$$
where
$$F\left(D_{k},C_{\ell }\right)=\frac{2\times \mathrm{Recall}\left(D_{k},C_{\ell }\right)\times \mathrm{Precision}\left(D_{k},C_{\ell }\right)}{\mathrm{Recall}\left(D_{k},C_{\ell }\right)+\mathrm{Precision}\left(D_{k},C_{\ell }\right)},$$
which is the harmonic mean of $\mathrm{Precision}(D_{k},C_{\ell })$ and $\mathrm{Recall}(D_{k},C_{\ell })$, and is called the F-measure [29].

The cluster level similarity between the estimated center $\hat{\mu }=\left({\hat{\mu }}_{1},\ldots ,{\hat{\mu }}_{K}\right)$ and the reference value (ground truth) of center $\mu^{*}=\left(\mu_{1}^{*},\ldots ,\mu_{K}^{*}\right)$ is the centroid index (CI) proposed by [26] as
$$\begin{array}{c} \mathrm{CI}\left(\hat{\mu },\mu^{*}\right)=\max\left({\mathrm{CI}}^{\prime }\left(\hat{\mu },\mu^{*}\right),{\mathrm{CI}}^{\prime }\left(\mu^{*},\hat{\mu }\right)\right), \end{array}$$
where
$$\begin{array}{ccc} {\mathrm{CI}}^{\prime }\left(\hat{\mu },\mu^{*}\right) & = & \sum_{k=1}^{K}\mathrm{orphan}\left(\mu_{k}^{*}\right)\\ \mathrm{orphan}(\mu_{k}^{*}) & = & \left\{\begin{array}{cc} 1 & q_{\ell }\neq k\;\forall \ell \\ 0 & \mathrm{otherwise} \end{array}\right.\\ q_{\ell } & = & \underset{1\leq k\leq K}{\mathrm{argmin}}{∥{\hat{\mu }}_{\ell }-\mu_{k}^{*}∥}^{2}. \end{array}$$
Here, $q_{\ell }$ indicates the index of the element of the reference center $\mu^{*}$ that is the nearest to ${\hat{\mu }}_{\ell }$. The function $\mathrm{orphan}(\mu_{k}^{*})$ indicates whether $\mu_{k}^{*}$ is an isolated element (orphan) or not, which is not nearest to any elements of $\hat{\mu }$. Hence ${\mathrm{CI}}^{\prime }\left(\hat{\mu },\mu^{*}\right)$ indicates the dissimilarity between $\hat{\mu }$ and $\mu^{*}$. Due to the asymmetry of ${\mathrm{CI}}^{\prime }\left(\hat{\mu },\mu^{*}\right)$ with respect to $\hat{\mu }$ and $\mu^{*}$, we take the maximum of ${\mathrm{CI}}^{\prime }\left(\hat{\mu },\mu^{*}\right)$ and ${\mathrm{CI}}^{\prime }\left(\mu^{*},\hat{\mu }\right)$. Hence $\mathrm{CI}(\hat{\mu },\mu^{*})$ measures how many clusters are differently located among $\hat{\mu }$ and $\mu^{*}$.

Another metric to measure the similarity between $\hat{\mu }$ and $\mu^{*}$ is defined as the mean squared error (MSE) over the number of clusters *K*, which is given as
$$\mathrm{MSE}=\frac{1}{K}\sum_{k=1}^{K}\left|\right|{\hat{\mu }}_{k}-\mu_{k}^{*}{\left|\right|}^{2}.$$
Differently from Purity and F-value, MSE can be calculated based on only estimated and reference centers $\hat{\mu }$ and $\mu^{*}$. This property is useful in a situation where the reference class sets $D_{1},\ldots ,D_{K}$ are difficult to determine but $\mu^{*}$ is easily identified. We use MSE in the simulation studies to evaluate the accuracy of $\hat{\mu }$ and Purity and F-value in the analysis of benchmark datasets.

#### 2.4.2. Simulation Studies

We generate samples according to the density function $p_{\tau ,\beta }\left(\theta^{*}\right)$ in (Equation 50) using the Metropolis-Hastings algorithm [30,31] as
(79)$$x∼Z_{\tau ,\beta }\left(\theta^{*}\right){\left\{\sum_{k=1}^{K}\pi_{k}^{*}w\left(x_{i},\mu_{k}^{*},\sum_{k}^{*}\right)\right\}}^{1+\beta },$$
where $\mu_{1}^{*}={\left(0,0\right)}^{⊤}$, $\mu_{2}^{*}=\left(5,5\right)$, $\mu_{3}^{*}={\left(-5,-5\right)}^{⊤}$, $\pi_{1}^{*}=0.5$, $\pi_{2}^{*}=0.2$, $\pi_{3}^{*}=0.3$ and
$$\sum_{1}^{*}=\left(\begin{array}{cc} 2 & -0.5\\ -0.5 & 1 \end{array}\right),\;\sum_{2}^{*}=\left(\begin{array}{cc} 1 & 0\\ 0 & 1 \end{array}\right),\;\sum_{3}^{*}=\left(\begin{array}{cc} 1 & 0\\ 0 & 1 \end{array}\right).$$

Figure 1 illustrates the perspective plots and contour plots for (τ,β)=(0.5,1),(0.5,0), (10, 1). The shape of $p_{\tau ,\beta }\left(\theta^{*}\right)$ varies according to the values of τ and β. The Gaussian mixture distribution corresponds with τ=0.5 and β=0 in panel (b). When β=1, the variance of each component increases and the contours connect with each other. On the other hand, for a large value of τ=10, the distribution shows high peaks around the centers. This indicates that $p_{\tau ,\beta }\left(\theta^{*}\right)$, including fuzzy c-means when τ→∞, has a quite different shape from he Gaussian mixture distribution. Other versions of the shapes are also illustrated in Appendix D. The performance of each method is evaluated by MSE based on 100 simulated samples with sample size n=3000.

#### 2.4.3. Benchmark Data Analysis

The performance of our proposed method is evaluated using benchmark datasets prepared by [32]. It includes a variety of datasets with low and high cluster overlap, various sample sizes, low and high dimensionalities and unbalanced cluster sizes. Hence, these datasets are suitable for clarifying the statistical performance of the clustering methods. In this setting, we compare the performance of k-means, fuzzy c-means, Gaussian, PAM, and Pareto clustering as well as the variants of Pareto clustering with several values of (τ,β) as explained in Appendix A. The characteristics of the benchmark datasets such as the sample sizes, the number of clusters, and dimensionality are summarized in Appendix A.

## 3. Results

Figure 2 illustrates the results of MSE in the simulation studies. Pareto clustering provides the best performance in panel (a), where the samples are generated by the underlying distribution $p_{0.5,1}\left(\theta^{*}\right)$ of Pareto clustering. The shape of the distribution is similar to Gaussian mixture; however, the variance of each component becomes larger and the contour lines are connected to each other as in panel (a) of Figure 1. On the other hand, in panel (c), the variance of each component becomes smaller and contour lines are completely separated. In the both cases, the performances of the Gaussian mixture are clearly degenerated. In the case of panel (b) in which the data are generated from the Gaussian mixture, the performances are comparable to each other, suggesting that k-means, fuzzy c-means, PAM, and Pareto clustering are robust to the underlying distributions to some extent.

In the benchmark data analysis, metrics of Purity, F-value, and CI are evaluated in Appendix A, where variance $\sum_{k}$ and mixing proportion $\pi_{k}$(k=1,…,K) in Pareto clustering are estimated. For the two-dimensional shape datasets such as Flame, Compound, D31, Aggregation, Jain, Pathbased, and Spiral in the upper rows of Appendix A, existing methods such as k-means, fuzzy c-means, PAM, and Gaussian mixture outperform our proposed methods. In high-dimensional data with d=1024 (Dim1024), k-means and a Gaussian mixture do not work well. Other methods achieved the maximum value (1) of Purity. For datasets with a large number of clusters, D31 (K=31) and A3 (K=50), PAM performs best. For datasets with large sample size of n≥5000 and a moderate number of clusters K=8,15, our proposed method performs best. As for the effect of τ, it barely affects the performance of our proposed method. On the other hand, β slightly affects the performance, resulting that the intermediates among Gaussian mixture, Pareto clustering, k-means, and fuzzy c-means, such as GP, GPKF${}_{1}$, GPKF${}_{10}$, and GPKF${}_{100}$, show relatively good performances as a whole. We observe similar tendencies regarding the F-value (Appendix A).

As for the CI, the values are relatively small for all methods, suggesting that cluster locations are properly estimated. However, some methods do not work for some datasets: Gaussian mixture for A3 (CI = 18), Birch1 (CI = 34) and Birch2 (CI = 49); k-means for D31 (CI = 7), Dim1024 (CI = 4), A3 (CI = 6), Birch1 (CI = 12) and Birch2 (CI = 23); and fuzzy c-means for D31 (CI = 5), A3 (CI = 7), Birch1 (CI = 18) and Birch2 (CI = 25). On the other hand, PAM and our proposed methods show stable results. The results, where $\sum_{k}$ and $\pi_{k}$ are not estimated and fix $\sum_{k}=I$ and $\pi_{k}=1/K$ in Pareto clustering, are shown in Appendix A.

## 4. Discussion

We propose a new clustering method based on the generalized energy function derived from the Kolmogorov–Nagumo average. The survival function used in the generalized energy function plays an important role to ensure the minimum consistency of the parameters, which is shown in Lemma 1 using the property of divergence $G(\mu^{*},\mu )$. We consider two examples of the survival function based on the Pareto and Fréchet distributions and show a connection among k-means, fuzzy c-means, and Gaussian mixture, leading to new methods that are intermediates among them. For the underlying distribution of our method in (Equation 50), we observe that k-means and fuzzy c-means do not have probabilistic interpretations because the corresponding underlying distributions become singular. We also propose an estimating algorithm for cluster locations, variances, and mixing proportions using the MM algorithm.

Simulation studies and benchmark data analysis show that intermediates among k-means, fuzzy c-means, and the Gaussian mixture perform well. This observation suggests that our proposed method has a wide range of applications in which k-means, fuzzy c-means, and the Gaussian mixture are used. For example, simultaneous deep learning and clustering [33] in which a deep neural network and k-means are jointly used, image segmentation using fuzzy c-means in a deep neural network [34], an application of fuzzy c-means in classification problems [35] and a parallel computation for large datasets by fuzzy c-means [36] can be investigated in the framework of the generalized energy function by the Pareto distribution.

As for the tuning parameters τ and β, we consider an approach using the leave-one-out cross validation in the Appendix A in order to improve the clustering performance. The objective function in the leave-one-out cross validation is derived from an anchor loss as in [37] to estimate the optimal values of τ and β properly. The benchmark data analysis suggests that the performance is insensitive to the values of τ but is sensitive to the values of β. Hence, this approach should be useful to determine the optimal value of β.

Banerjee et al. [38] has proposed a clustering method based on Bregman divergences and clarified the relationship between the exponential families and the corresponding Bregman divergences. They separately consider hard and soft clustering; the former corresponds to k-means style clustering and the latter corresponds to mixture model clustering. In our proposed model, the tuning parameters τ and β bridge the gap between them and the performances are investigated by simulation studies and benchmark datasets. The extension of our method by replacing the squared distance $∥x_{i}-\mu_{k}{∥}_{\sum_{k}^{-1}}^{2}$ with Bregman divergence should improve its practical flexibility and utility. When β or γ divergence is used, the clustering method should be robust to contamination in observations as suggested by [39,40].

It is well known that MM algorithm and EM algorithm converge to a local optimum and the resultant clusters are sensitive to initial values [41]. One way to circumvent this difficulty is to prepare several sets of initial values and select the best one among them such as the global k-means algorithm [42]. Another approach is to combine MM algorithm and genetic algorithm (GA) to expand thoroughly the search space for the optimal solution [41,43]. The both approach can be incorporated into the Pareto clustering to make it robust to the initial values and to escape from local optimal solutions.

## Figures and Tables

**Figure 1 entropy-23-00518-f001:**
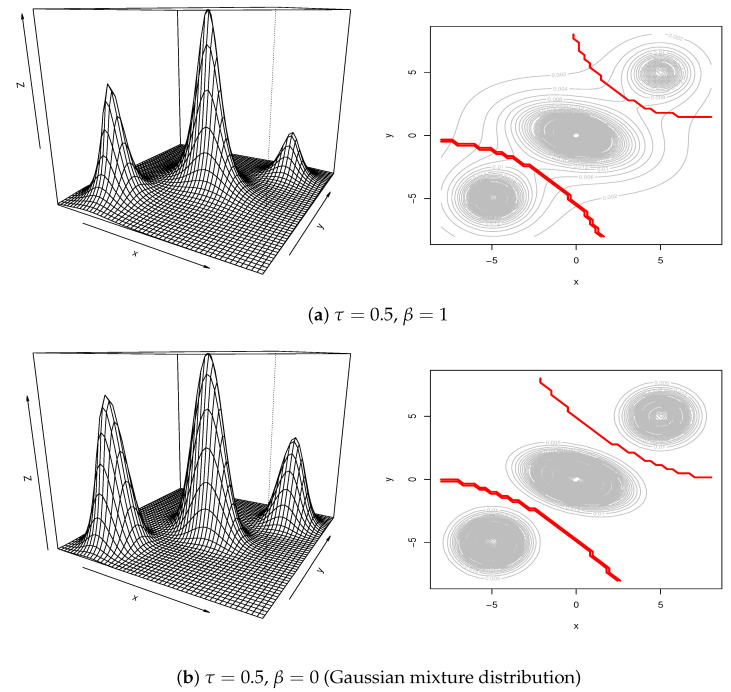
Perspective plots (left panels) and contour plots for $p_{\tau ,\beta }\left(\theta^{*}\right)$ with boundaries marked in red for (**a**) τ=0.5, β=1, (**b**) τ=0.5, β=0 and (**c**) τ=10, β=1.

**Figure 2 entropy-23-00518-f002:**
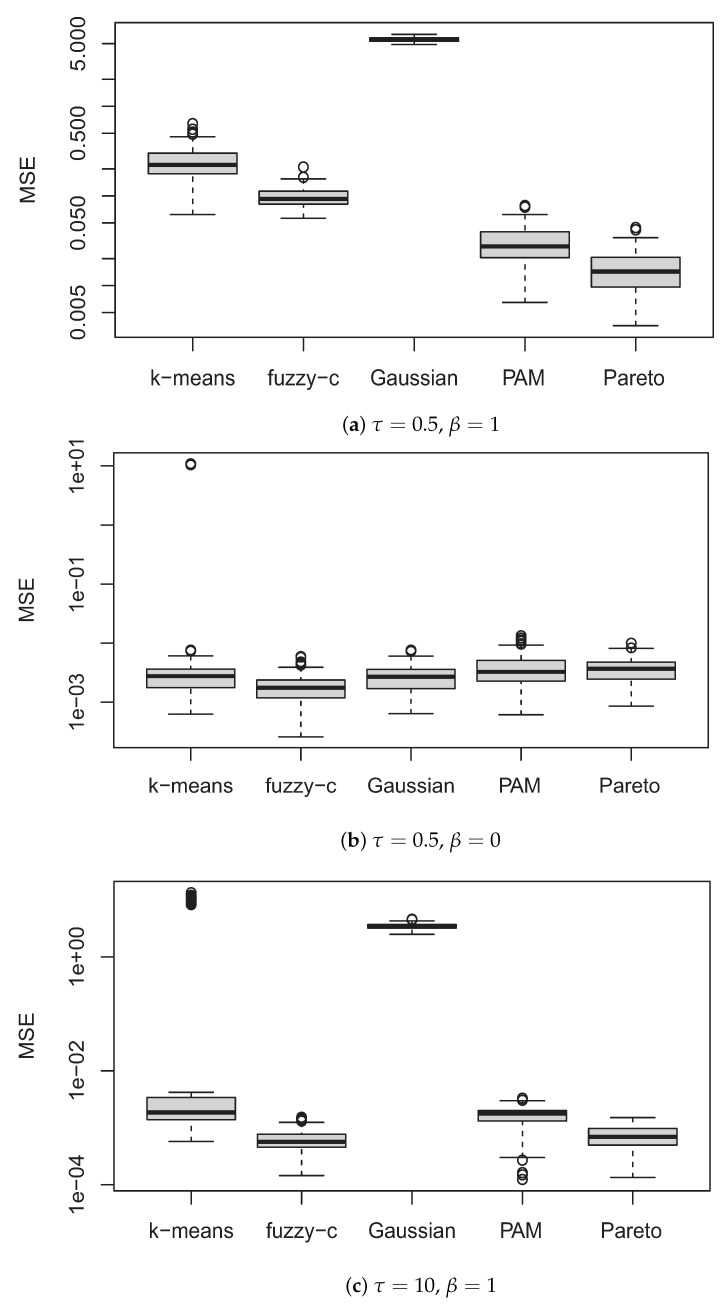
Mean squared errors (MSE) on the log scale based on 100 random samples for each method. The samples are generated based on $p_{\tau ,\beta }\left(\theta^{*}\right)$ with (**a**) τ=0.5, β=1, (**b**) τ=0.5, β=0 and (**c**) τ=10, β=1.

## Data Availability

Benchmark datasets are available at http://cs.uef.fi/sipu/datasets/.

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
