# Peer review of "A Unified Formulation of k-Means, Fuzzy c-Means and Gaussian Mixture Model by the Kolmogorov–Nagumo Average"

_entropy, 2021, doi:10.3390/e23050518_

Round 1

Reviewer 1 Report

This paper has proposed a new clustering method based on the generalized energy function derived from the Kolmogorov-Nagumo average. It also connects together a few famous clustering methods and provides a unified understanding for them. Overall, this paper is quite interesting and it provides insightful viewpoints. I have only a few minor comments listed below.

  • p.2 line 53 and some other places: survivor function -> survival function
  • p.2 line 54: an generalized -> a generalized
  • p.15 line 272: This observations -> This observation
  • In all you data analysis, the number of clusters, K, is assumed known?
  • Supplementary: For dataset "Dim1024" and with the case Sigma not equal to identity, how did you handle the Sigma inverse in your data analysis? High dimensional covariance estimation is quite challenging and the estimation of its inverse is even more challenging.

Reviewer 2 Report

In the paper, a novel clustering based on the Kolmogorov-Nagumo average is introduced. The proposed approach is interesting and the presented findings seem valid. However, the following comments should be addressed:
1.    The title of the paper is somewhat misleading. The study does not deepen understanding of used clustering methods or their common properties. (Even the word ‘understanding’ is used only in the title.) This should be clarified. The joint usage of several clustering methods can be seen as a unification approach.  However, it is not explicitly elaborated.
2.    The contributions of this paper should be enlisted at the end of Section 1.
3.    Are there other Pareto clustering approaches in the literature? They should be described and compared, e.g., “A Cluster Truncated Pareto Distribution and Its Applications” https://doi.org/10.1155/2013/265373   
4.    The introduced clustering incorporates other clustering types. Please provide a discussion, supported by an experiment, which clustering method contributes the most to its performance. Would its results be improved with the addition of more methods? How far are we from the optimal clustering?
5.    To facilitate the analysis, it is advised to present all the tables in the main text instead of the supporting information. It is inconvenient to jump there to see the benchmark datasets and back to Section 3 and there again to see the outputs. 
6.    The reported experiments cannot be replicated. Please share the source code with readers, ensuring the reproducibility of the results (mandatory).  This would greatly promote the paper.

Reviewer 3 Report

see attachment

Round 2

Reviewer 2 Report

The revised manuscript addresses all identified issues.

Author Response

Thank you for your useful comments and suggestions.

Reviewer 3 Report

attached
